# Continuous Monitoring of Entropy Production and Entropy Flow in Humans Exercising under Heat Stress

**DOI:** 10.3390/e25091290

**Published:** 2023-09-03

**Authors:** Nicolas Brodeur, Sean R. Notley, Glen P. Kenny, André Longtin, Andrew J. E. Seely

**Affiliations:** 1Department of Physics, Faculty of Science, University of Ottawa, Ottawa, ON K1N 6N5, Canada; nbrod033@uottawa.ca (N.B.);; 2Human and Environmental Physiology Research Unit, Faculty of Health Sciences, University of Ottawa, Ottawa, ON K1N 6N5, Canada; 3Faculty of Medicine, University of Ottawa, Ottawa, ON K1H 8M5, Canada; 4Ottawa Hospital Research Institute, Ottawa, ON K1Y 4E9, Canada

**Keywords:** entropy production, entropy monitoring, calorimetry, heat flow, exercise stress

## Abstract

Complex living systems, such as the human organism, are characterized by their self-organized and dissipative behaviors, where irreversible processes continuously produce entropy internally and export it to the environment; however, a means by which to measure human entropy production and entropy flow over time is not well-studied. In this article, we leverage prior experimental data to introduce an experimental approach for the continuous measurement of external entropy flow (released to the environment) and internal entropy production (within the body), using direct and indirect calorimetry, respectively, for humans exercising under heat stress. Direct calorimetry, performed with a whole-body modified Snellen calorimeter, was used to measure the external heat dissipation from the change in temperature and relative humidity between the air outflow and inflow, from which was derived the rates of entropy flow of the body. Indirect calorimetry, which measures oxygen consumption and carbon dioxide production from inspired and expired gases, was used to monitor internal entropy production. A two-compartment entropy flow model was used to calculate the rates of internal entropy production and external entropy flow for 11 middle-aged men during a schedule of alternating exercise and resting bouts at a fixed metabolic heat production rate. We measured a resting internal entropy production rate of (0.18 ± 0.01) W/(K·m^2^) during heat stress only, which is in agreement with published measurements. This research introduces an approach for the real-time monitoring of entropy production and entropy flow in humans, and aims for an improved understanding of human health and illness based on non-equilibrium thermodynamics.

## 1. Introduction

Physical systems generally evolve towards a state of equilibrium characterized by complete homogeneity which is, in essence, a manifestation of the second law of thermodynamics and its associated principle of entropy maximization. However, the emergence and evolution of life forms on Earth shows the opposite trend; from the assembly of the first amino acids to the formation of the first bacterium, up to the first plants and mammals, it turns out that at each stage of development life is found to be more ordered and, thus, more unlikely. Then how can life elude the second law of thermodynamics? In fact, non-equilibrium systems, like cells, organs, and organisms, actively exchange matter and energy with their environment. Far from a state of equilibrium, new structures and functions can emerge from the complex interplay of irreversible processes that continuously produce entropy as energy gradients are dissipated. These systems, referred to as dissipative structures, are characterized by self-organized behavior in the presence of sufficiently strong energy gradients. In such cases, complex systems can spontaneously form to feed upon these energy gradients, adopting highly ordered configurations that are impossible under equilibrium conditions. Complex living systems, such as the human body, can be regarded as open thermodynamical machines that inherently transform energy sources, mostly into heat, in order to produce a relatively small portion of useful work. Here, we develop an experimental–theoretical framework to understand heat and entropy flows in healthy humans who alternate work and rest periods.

The rate of entropy change in a system can be expressed as the sum of the rates of entropy produced within the system and the entropy exchanged with its surroundings. Mathematically put, the entropy balance equation can be written as
(1)$$\frac{dS}{dt}=\frac{dS_{i}}{dt}+\frac{dS_{e}}{dt}$$
where dS/dt corresponds to the rate of entropy change of the system, $dS_{i}/dt$ is the internal entropy production, and $dS_{e}/dt$ is the entropy exchange (or entropy flow) with the environment through the system’s boundaries. We note that this entropy balance equation for a system is compatible with the increase in entropy in the universe when irreversible processes take place. Irrespective of the absolute value and sign of dS/dt, the rate of entropy change of the universe will always be positive. It is well known from Prigogine’s work that for near-equilibrium steady states, for which linear relationships between the thermodynamic forces and fluxes are assumed, entropy production is minimal [1]. Intuitively, if some set of constraints prevents a system from achieving an equilibrium state where entropy production is zero, then the closest state to equilibrium becomes the single steady state where entropy production is minimal. This principle generally applies to linear systems that exhibit only a few degrees of freedom [2].

The application of non-equilibrium thermodynamics to biophysical systems has a long-standing history [3]; however, it has been traditionally restricted to microscopic considerations down to the level of chemical reactions. On a larger scale, Zakharov and Sadovsky developed a theoretical model for the thermal regulation of animals based on the entropy production principle [4], but their analysis focuses on passive heat exchange with the environment (i.e., heat conduction and diffusion), which is not the main mechanism through which humans dissipate heat during exercise (i.e., skin cooling from the evaporation of sweat). On the experimental side, Aoki studied the entropy flows and entropy production of the human body under basal conditions at different ambient temperatures using the calorimetry measurements of Hardy and Du Bois [5,6]. His calculations included contributions from the entropy flow of energy exchange (i.e., heat and radiation) and mass exchange associated with the respiration process, although the latter was found to be negligeable compared to the former. Others have studied the entropy generation of humans during their lifespan [7], while a general interest is growing to link thermodynamics and entropy considerations with health and disease [8,9]. Overall, there exist a very limited number of studies that provide experimental investigations of the production and exchange of entropy in complex living systems and none, to our knowledge, that can monitor entropy production in real-time.

Entropy, unlike energy, is not a conserved quantity; rather, entropy is preserved in that, once created, it cannot be destroyed [10]. It can, however, be dissipated externally from the system that created it to prevent entropy accumulation within. In this sense, entropy production refers to the irreversible transformation of energy within a system (e.g., metabolic activities), while entropy flow refers to the rate of transfer of entropy across a boundary (e.g., from the skin to the ambient room).

Stationary states resemble equilibrium states in that their thermodynamic properties, like temperature and entropy, do not vary over time. However, unlike in equilibrium, heat and entropy flows can occur in a stationary state, while maintaining a constant temperature, provided that the thermodynamic flows that enter the system or are created within it, are exactly matched by their outflowing counterparts. In the case of living systems, stationarity is obtained when the net rate of entropy change in the body is zero; in that case, the internal entropy produced by metabolic irreversible processes is exactly balanced by the entropy dissipated to the environment. However, the human body does not have an infinitely fast reaction time, such that any increase in internal entropy production will be immediately matched by an equal increase in entropy flow to the environment. In other words, there will be a transient period, for example after the onset of exercise, when the body is not in a stationary state, which leads to an increase in body temperature and the accumulation of entropy. Although the body can be considered to be approximately at a steady state for the timescale of a day [5], participants are not expected to be in a steady state for the duration of an experiment where heat stress and exercise are involved [11].

In the present paper, we introduce a two-compartment entropy flow model for the continuous monitoring of entropy production in humans evaluated during physical exercise under heat stress. The thermodynamical definition of entropy will be applied to this experiment, as opposed to the informational definition of entropy derived from information theory. The system under study here, namely the human body, is considered a classical system, not a quantum one, since it cannot be properly described in terms of quantized microstates. Although it remains debated whether the entropy of the human body can be measured, entropy rates and their imbalances are quantifiable and measurable.

## 2. Materials and Methods

### 2.1. Human Subjects and Experimental Design

The results of this experiment are based on prior published data which studied the impact of heat and exercise stress on humans. The experimental design of the study has been previously and thoroughly described [12]. Briefly, heat transfers were measured during exercise and resting periods under heat stress for 11 healthy and habitually active middle-aged males (43 ± 2 years). In the study, participants entered the calorimetry chamber and initially rested for 30 min. Then, they performed four 15 min exercise bouts of cycling on an upright, seated cycle ergometer, at a constant rate of metabolic heat production equal to 400 W. Each bout was separated by a 15 min resting period with a final recovery period of 60 min. Physiologic properties, such as body temperature and heart rate, as well as thermodynamic properties, such as heat and entropy flows, were continuously monitored and reported as averages over 1 min intervals. The temperature inside the calorimetry chamber was set to 35 °C with a relative humidity of 20%. The calorimetry chamber thus essentially acts as a heat bath at a constant temperature and relative humidity by imposing fixed boundary conditions on the participants.

### 2.2. Internal Heat Production Measured with Indirect Calorimetry

The modified Snellen calorimeter, shown in Figure 1, is a state-of-the-art whole-body air calorimeter that allows the study of human heat exchange in different ambient conditions [13]. It provides a very precise, continuous measure of the heat dissipated (dry ± evaporative heat exchange) by the human body during rest and exercise [14]. When combined with the rate of internal heat production, or metabolic heat production (indirect calorimetry), body heat storage can be quantified. 


Internal heat production is derived from the metabolism. The chemical energy stored within the body and liberated through metabolic processes is transformed into external work (e.g., cycling) and metabolic heat (see Figure 2b). Written in the form of the First Law of Thermodynamics, the rate of change in the internal chemical energy corresponds approximately to the metabolic energy expenditure, $\dot{M}$ (i.e., chemical energy liberated), and is given by
(2)$$\frac{dU}{dt}\approx \dot{M}={\dot{Q}}_{int}+\dot{W}$$
where ${\dot{Q}}_{int}$ is the rate of metabolic heat production and $\dot{W}$ is the external work rate performed by the participants on the cycle ergometer. We assume that the internal work performed by the organs to generate internal flows (e.g., the heart pumping blood) is ultimately transformed into heat through friction and dissipation, which is accounted for in the measurement of metabolic heat production, because chemical energy is needed to provide this work. The metabolic energy expenditure, $\dot{M}$, can be measured using indirect calorimetry and the external work rate, $\dot{W}$, is known; the metabolic heat production, ${\dot{Q}}_{int}$, can then be readily obtained from Equation (2)
(3)$${\dot{Q}}_{int}=\dot{M}-\dot{W}.$$The rate of metabolic energy expenditure, $\dot{M}$, was estimated spirometrically from the respiratory exchange ratio, *R*, between the rate of carbon dioxide production, ${\dot{V}}_{CO_{2}}$, and the rate of oxygen consumption, ${\dot{V}}_{O_{2}}$ [14], both measured at L·min^−1^, using the following equation
(4)$$\dot{M}=\frac{\left({\dot{V}}_{O_{2}}\cdot \left[\frac{R-0.7}{0.3}e_{c}+\frac{1-R}{0.3}e_{f}\right]\right)}{60}\;$$
where $e_{c}$ is the caloric equivalent per liter of oxygen for the oxidation of carbohydrates ($e_{c}=21,130\;$J), and $e_{f}$ is the caloric equivalent per liter of oxygen for the oxidation of fat ($e_{f}=19,630\;$J). The value for R is measured in real-time throughout the experiment to calculate (and regulate)heat production. An R value near 0.7 indicates that fat is the predominant fuel source, a value of 1.0 is indicative of carbohydrates being the predominant fuel source, and a value between 0.7 and 1.0 suggests a mix of both fat and carbohydrates. The expired air was directed into a mixing box where ${\dot{V}}_{O_{2}}$ was measured and then vented back into the calorimetry chamber, so that temperature and humidity differences between the chamber and respiration gases could be accounted for. The external work rate, $\dot{W}$, was continuously adjusted by changing the cycling resistance during the exercise periods to ensure the targeted metabolic heat production, ${\dot{Q}}_{int}$, of 400 W remained constant over time. The typical values for the work rates were in the range of 70 W (not part of the 400 W target for ${\dot{Q}}_{int}$), which represents moderate-to-high intensity exercise. Historically, indirect calorimetry has been regarded as the gold standard and still is the reference standard and clinically recommended mean for the accurate measurement of energy expenditure [15,16]; alternative methods can approximate $\dot{M}$ and ${\dot{Q}}_{int}$ from the heart rate, subjective sensations, or empirical tables [14].

### 2.3. External Heat Dissipation Measured with Direct Calorimetry

The rate of external heat transfer from the body to the surroundings, denoted by ${\dot{Q}}_{out}$, corresponds to the sum of dry heat loss, ${\dot{Q}}_{dry},$ evaporative heat loss, ${\dot{Q}}_{evap}$, and heat loss through respiration, ${\dot{Q}}_{resp}$, such that
(5)$${\dot{Q}}_{out}={\dot{Q}}_{dry}+{\dot{Q}}_{evap}+{\dot{Q}}_{resp}\;$$

Dry heat loss (${\dot{Q}}_{dry}$) results from heat exchange with the environment via conduction, convection, and radiation at the skin surface, and is given by
(6)$${\dot{Q}}_{dry}=c_{air}\times {\dot{m}}_{air}\times \Delta T_{air}$$
where $c_{air}=1.005\;J\cdot {\left(kg\cdot {}^{\circ}C\right)}^{-1}$ is the specific heat of air, ${\dot{m}}_{air}$ is the mass flow of air (kg air/s) out of the calorimetry chamber, and $\Delta T_{air}$ is the difference in temperature between the outflow and inflow of air from the calorimetry chamber. The evaporative heat loss, ${\dot{Q}}_{evap}$, consists of the heat dissipated from the skin resulting from the evaporation of sweat, and is calculated from the change in absolute humidity inside the calorimeter
(7)$${\dot{Q}}_{evap}=L_{vap}\times {\dot{m}}_{air}\times \Delta \rho_{v}$$
where $L_{vap}=2.426\;J$ per gram of sweat and is the latent heat of the vaporization of sweat, and $\Delta \rho_{v}$ is the difference in absolute humidity (g of water/kg air) between the outflow and inflow of air from the calorimetry chamber. The modified Snellen whole-body air calorimeter has an accuracy of ±2.3 W for the measurement of total body heat loss, ${\dot{Q}}_{out}$, representing a measurement error that is smaller than 1% [13]. Lastly, the heat loss through respiration, ${\dot{Q}}_{resp}$, results from the dry and evaporative heat transfer between respired gases and the body. In the current experimental setup, the respired gases were vented back into the calorimetry chamber and thus were mixed with the air outflow, from which were measured ${\dot{Q}}_{dry}$ and ${\dot{Q}}_{evap}$ [12]. Although it only had a negligeable contribution to the total heat loss because the ambient temperature was very close to body temperature at 35 °C, ${\dot{Q}}_{resp}$ was indirectly accounted for through the measurement of ${\dot{Q}}_{dry}$ and ${\dot{Q}}_{evap}$.

### 2.4. Rate of Heat Storage in the Body

The heat produced internally (i.e., metabolic heat production, ${\dot{Q}}_{in}$) is partly dissipated to the environment as ${\dot{Q}}_{out}$, and partly stored within the body; the balance between the rate of metabolic heat production and the rate of heat dissipation corresponds to the rate of heat storage: (8)$${\dot{Q}}_{st}={\dot{Q}}_{in}-{\dot{Q}}_{out}\;$$Heat stored within the body leads to increase in body temperature, and thus the rate of heat storage can serve to calculate variations in body temperature, as described in the following section. Additionally, note that energy can also be stored within the body through the formation of energy-rich compounds. However, as the experiment of Larose et al. shows, energy storage can be neglected as the participants had a light meal in the morning and no subsequent food intake throughout the duration of the experiment [12].

### 2.5. Temperature Measurements Using Thermometry and Calorimetry

Thermometry was used to measure the core and skin temperatures. The core temperature was measured by inserting a rectal temperature probe a minimum of 12 cm past the anal sphincter. The skin temperature was calculated using a weighted average of four skin temperature probes located on the upper back (30%), chest (30%), quadriceps (20%), and back calf (20%). Then, the whole-body temperature was estimated via a weighted average of the core temperature (90%, measured with the rectal probe) and skin temperature (10%, measured by the weighted average of the four different skin probes).

While thermometry provides accurate temperature measurements under resting conditions, it tends to underestimate temperature variations during exercise [17]. Hence, the baseline values for the core temperature were determined using thermometry during the initial resting period. During the subsequent exercise and recovery periods, the relative changes in body temperature, $∆T_{b}$, during a time interval, ∆t, were determined using calorimetry through the balance of heat flows, and are given by
(9)$$∆T_{b}=\frac{{\dot{Q}}_{st}∆t}{m\cdot c_{p}}$$
where ${\dot{Q}}_{st}$ corresponds to the rate of heat storage in the body, defined using the heat balance given by Equation (8). The numerator of Equation (9) corresponds to the change in body heat content during the time interval, ∆t, m is the total body mass of the participant (in kg), and $c_{p}$ is the specific heat capacity of living tissues (in J·kg^−1^·K^−1^). 

### 2.6. Two-Compartment Non-Stationary Model of Entropy Production 

In a multicompartment model, the rate of entropy change of an open system is given by the sum of the rates of entropy change within each subsystem, k (see Figure 2). Using a two-compartment core–skin model, the rate of entropy change of the body is given by
(10)$$\left(\frac{dS}{dt}\right)=\sum_{k=1}^{n}{\left(\frac{dS}{dt}\right)}_{k}={\left(\frac{dS}{dt}\right)}_{core}+{\left(\frac{dS}{dt}\right)}_{skin}$$
where the local rate of entropy change within subsystem *k* is given by
(11)$${\left(\frac{dS}{dt}\right)}_{k}=-\frac{1}{T_{k}}\vec{\nabla }\cdot {\vec{J}}_{k}+\sigma_{k}.$$The first term on the right-hand side of Equation (11) corresponds to the divergence of the total heat flow, ${\vec{J}}_{k}$, through the boundary of subsystem *k,* and the second term, $\sigma_{k},$ corresponds to the entropy production due to irreversible processes occurring within subsystem *k*, which must be greater or equal to zero according to the Second Law of Thermodynamics ($\sigma_{k}\geq 0$). We determined that the temperature variations between two consecutive data recordings a minute apart were very small; therefore, we could neglect any term containing time derivatives of temperature. We restricted the entropy-rate contributions to real-time heat production and dissipation because the human body is physiologically inefficient at converting chemical energy into mechanical work (e.g., lifting a box, cycling, etc.). Depending on the task performed, between 80% and 100% of the energy is converted into heat along the metabolic pathways [14]. Further, Aoki showed that the entropy flow associated with the mass–flow of respiratory gases can be neglected [1].

The internal entropy production within the core is driven mainly by heat production from metabolic activities, i.e., $\sigma_{c}={\dot{Q}}_{int}/T_{c}$. It is important to note that indirect calorimetry, used to estimate the metabolic heat production through gas exchange, cannot determine the oxygen consumption in each compartment because it is a whole-body measurement. Therefore, the measurement of metabolic heat production also includes the heat generated by the skin. Since entropy production from heat generation within the skin is thus already accounted for, we can assume that $\sigma_{s}=0$. Equation (10) then becomes:(12)$$\left(\frac{dS}{dt}\right)=\frac{{\dot{Q}}_{int}}{T_{c}}+{\dot{Q}}_{t}\left(\frac{1}{T_{s}}-\frac{1}{T_{c}}\right)-\frac{{\dot{Q}}_{out}}{T_{s}},$$
where ${\dot{Q}}_{t}$ is the heat flow transferred from the core to the skin, $T_{c}$ and $T_{s}$ are, respectively, the core and skin temperatures, ${\dot{Q}}_{int}$ is the rate of metabolic heat production in the body, and ${\dot{Q}}_{out}$ is the rate of heat dissipation from the skin to the environment. Equation (12) can be derived similarly from the multi-box model used by Ozawa et al. in the context of entropy production in the planetary atmosphere [18].

The second term of Equation (12) corresponds to the entropy change associated with the transfer of heat from two compartments at different temperatures (i.e., core and skin). If the core and skin temperatures were equal, this term would vanish as expected. Most importantly, Equation (12) is valid for both stationary and non-stationary states. In stationary states, body heat content and temperatures are time-independent, implying that all heat flows are equal, which consequently leads to dS/dt=0.

The heat flow transferred from the core to the skin, ${\dot{Q}}_{t}$, cannot be evaluated with the present experimental setup. If we define $\Delta T=T_{c}-T_{s}$ as the difference between core and skin temperatures, we find that
(13)$$\left(\frac{1}{T_{s}}-\frac{1}{T_{c}}\right)\approx \frac{\Delta T}{T_{s}^{2}}$$
provided that Δ*T* is small. Indeed, the relative difference between the core and skin temperature is close to 1% when expressed in Kelvin units. This justifies the following approximation
(14)$${\left(\frac{dS}{dt}\right)}_{body}=\frac{dS_{i}}{dt}+\frac{dS_{e}}{dt}\approx \frac{{\dot{Q}}_{int}}{T_{c}}-\frac{{\dot{Q}}_{out}}{T_{s}}$$It can be seen from Equation (14) that the rate of entropy production is given by $dS_{i}/dt={\dot{Q}}_{int}/T_{c},\;$and the rate of entropy flow is given by $dS_{e}/dt={-\dot{Q}}_{out}/T_{s}$. It is important to note that even though we use the standard notation for time derivatives, expressed in J/K per second, the heat and temperature measurements are reported as averages over 1 min intervals. Hence, entropy rates in units of (J·K^−1^·s^−1^) are obtained by converting heat measurements from 1 min intervals (i.e., per minute) into per second intervals. The current experimental setup does not have a time resolution down to the second, but it has an incredible accuracy for heat measurements on the order of one minute [13].

Finally, the cumulative entropy change in the body,$\;\Delta S\left(t\right)$, which can be interpreted as a measure of thermodynamic irreversibility or stress, is calculated as
(15)$$\Delta S\left(t\right)=\int_{t_{0}}^{t}\left(\frac{dS}{{dt}^{\prime }}\right)dt^{\prime }\approx \sum_{i=0}^{t}{\langle \frac{dS}{dt}\rangle }_{i}\cdot \Delta t_{i}$$
where the rates of entropy change are averaged over 1 min intervals.

### 2.7. Data and Statistical Analysis

The data analysis was performed using an in-house program written in MATLAB R2023a (The MathWorks, Natick, MA, USA). When any normalization was applied to the time-series curves, the values were normalized for each individual prior to calculating the group averages. Unless stated otherwise, all the entropy rates and changes presented below were normalized using unit body mass using the participants’ weight.

## 3. Results

### 3.1. Resting Entropy Production during Heat Stress 

The resting rates of entropy production during heat stress, shown in Table 1, were calculated by averaging the entropy production rates during the initial resting period inside the calorimetry chamber and normalized using body-surface area (BSA) or mass. The BSA was estimated from the measures of standing height and body mass following the standard method of Du Bois and Du Bois [19]. The BSA-normalized value for the resting entropy production rate is in good agreement with that of Aoki [5], while our mass-normalized value is approximately 25% higher. This discrepancy can be easily understood by considering the differences in the experimental design. In Aoki’s study, the data were recorded for a single individual who was 54 years old, 179 cm in height, 74.7 kg in weight, and had an estimated BSA of 1.54 m^2^. Aoki also showed that the production of entropy under basal conditions is nearly constant for calorimeter temperatures in the range of 26–32 °C, with an average basal rate of specific entropy production (i.e., per unit body area) of 0.172 ± 0.003 W/(K⋅m^2^). In contrast, our study involved 11 participants with an average age of 43 ± 2 years, weight of 84 ± 6 kg, and an estimated BSA of 2.0 ± 0.1 m^2^. Moreover, the ambient calorimeter temperature in our study was set to 35 °C, which is outside the range investigated by Aoki. We could not identify other independent measurements of human entropy production to which we could compare our results.

### 3.2. Entropy Rate Curves

The rates for internal entropy production, external entropy flow and their difference, and whole-body entropy change are shown in Figure 3. Each entropy rate curve was normalized using the individuals’ weights prior to calculating the group averages. Figure 3a shows the group-averaged external entropy production rate, $dS_{i}/dt$ (in blue), and the absolute values of entropy flow, $dS_{e}/dt$ (in red), with their respective shaded regions representing plus/minus one standard deviation. The entropy production rates for the first 30 min represent the resting entropy production rates. The initial rest period was followed by four periods of exercise, during which the production of metabolic heat was fixed and maintained at a constant level by the experimental design. The entropy flows initially rose at the start of exercise, although at a slower rate than the entropy production rates; they also declined faster than their initial rise at the start of the resting periods. The entropy flows did not recover their initial resting values over the course of the experiment, as seen in the higher plateaus during the inter-exercise resting periods. An additional cooldown time at the end of the experiment would have expectedly restored the entropy flows to their resting values. Moreover, the maximal entropy flows for the first exercise bout consistently reached a smaller value than those for the remaining three exercise bouts.

The rate of internal entropy change in the body is shown in Figure 3b; that is, the difference between the rate of entropy production and entropy flow plotted in Figure 3a. It should be noted that dS/dt is not restricted to positive values like the entropy production rate is; it can take positive or negative values, depending on the magnitude of both contributions and the sign of the entropy flows. In the first few minutes of exercise, the entropy production rate increases abruptly while the entropy flow increases almost linearly at a much lower rate. The difference between both terms thus takes a positive value. In contrast, at the onset of a resting period following an exercise bout, the entropy production rate decreases abruptly while the entropy flow is still elevated, which leads to a negative rate of entropy change. Entropy production and entropy flow are independent variables that originate from different mechanisms. While entropy production is most often activated consciously in living systems in response to an external stimulus (e.g., the cue to start exercising), entropy flow is an involuntary physiological response to the variation in internal entropy production or a notable change in ambient conditions.

Finally, Figure 3c shows the accumulation of entropy within the body, which corresponds to the time integration of the rate of entropy change from Figure 3b. The initial rise in entropy accumulation is due to the lack of entropy balance when the participants entered the calorimetry chamber. Indeed, it typically takes multiple hours for participants to achieve heat balance when resting under such conditions [14,20]. Each exercise bout is associated with a significant increase in accumulated entropy, followed by a smaller decrease during the resting periods. We noticed that the entropy accumulated over the experiment did not completely dissipate despite the recovery period of 60 min at the end. Instead, a new stationary state was found to exist, where entropy production and entropy flow were approximately balanced. The net rate of entropy change vanished, and thus the entropy was no longer accumulating, but instead steady around an asymptotic value that reflects the net change in body entropy following the recovery period.

### 3.3. Quasi-Static Entropy Change Model

The variation in body temperature over a short period of time (e.g., 1 min) is relatively small; in fact, it is sufficiently small that the slow rate of change in body temperature could be considered as the result of an internal quasi-static process. Fundamental in classical thermodynamics, a quasi-static process is an idealized process in which a system undergoes incremental and slow changes such that it is, at every instant, at equilibrium [21]. Time series can thus be viewed as a succession of equilibrium states that are infinitely close to one another. In our experiment, heat storage within the body increased incrementally in a similar manner during the exercise periods. Using the well-known formalism of classical thermodynamics, the infinitesimal entropy change associated with the transfer of heat, dQ, is given by dS=dQ/T. The heat transfer, dQ, can be expressed as a temperature variation, dT, through dQ=mc⋅dT, with m being the mass of the participant, and c being the average specific heat of the body. Therefore, the entropy change within the body over a single interval is given by
(16)$$\Delta S_{i}=\int_{T_{i}}^{T_{i+1}\;}\frac{dQ}{T}=mc\times ln\left(\frac{T_{i+1}}{T_{i}}\right)$$
where $T_{i}$ and $T_{i+1}$ are, respectively, the initial and final temperatures over a 1 min interval. These temperatures can be estimated using either thermometry (i.e., the use of temperature probes) or calorimetry (i.e., the balance of heat flows) [14]. Since quasi-static processes are akin to a succession of equilibrium states, no variations in time (or rates) are formally defined. In other words, all that can be analyzed is the entropy change during a single interval, and not the instantaneous rates that typically appear as time derivatives.

Figure 4 shows the entropy change in the body calculated from Equation (16) using two distinct methods for core temperature measurement, namely, thermometry (in red) and calorimetry (in blue). Thermometry was found to provide a noisy time series for entropy change in the body. The higher variance for each data point compared to the calorimetry time series originates from the inability of the rectal probes to adequately assess changes in core temperature. We have also found that thermometry appears to underestimate the entropy change at critical points, like the onset and offset of exercise periods, as observed in Figure 4, while the calorimetry time series exhibits a larger amplitude between the extrema. This observation also suggests that thermometry has a delayed reaction compared to the calorimetric approach. This delayed reaction to increases in entropy changes in the body can be attributed to a slower response to in temperature changes, which in turn is due to the low heat conductivity of the air surrounding the rectal probe [14,22]. Overall, the results suggest that thermometry is less-sensitive and -responsive compared to calorimetry for the continuous measurement of body entropy changes over short intervals. 

It should be noted that there is a negligible difference in the resulting rate of entropy change if the core temperature is measured via thermometry or calorimetry when using Equation (14). Nevertheless, the heat flows, ${\dot{Q}}_{int}$ and ${\dot{Q}}_{out}$, in the numerators of Equation (14) must be measured using calorimetry (indirect and direct calorimetry, respectively). Therefore, calorimetry remains essential for the calculation of entropy rates. However, the results shown in Figure 4 indicate a noticeable difference between both methods when using the quasi-static model when entropy change is not calculated from heat flows, but rather from temperature variations. In this case, calorimetry provides a much more reliable measurement than thermometry.

There is a striking similarity between the quasi-static approach that uses calorimetry to estimate temperatures and the two-compartment model that uses the balance of entropy flows (see the entropy change curve in Figure 3b). In fact, considering that the temperature variation, $\Delta T_{i}=T_{i+1}-T_{i}$, during a single interval is relatively small, one can expand the logarithm in Equation (16) into a first-order Taylor expansion, such that the entropy change in the body can be expressed as
(17)$$\Delta S_{i}=mc\left(\frac{\Delta T_{i}}{T_{i}}+\ldots \right)\approx \frac{mc\Delta T_{i}}{T_{i}}=\frac{{\dot{Q}}_{st}}{T_{i}}=\frac{{\dot{Q}}_{int}}{T_{i}}-\frac{{\dot{Q}}_{out}}{T_{i}}$$

The resemblance between Equations (14) and (17) is not surprising given that similar assumptions are made in their respective derivation. For example, small temperature variations, $\Delta T_{i}$, are implied in the entropy flow model by neglecting terms of higher order when taking the time derivative of temperature. Although these models lead to similar results, there are nevertheless key conceptual differences that allow for alternative interpretations. The main difference between both models relates to their thermodynamic interpretation. The quasi-static approach implies that the body is always near the equilibrium state (infinitely close), which is invalid in the case of living systems where equilibrium conditions are only obtained after death. On the other hand, the two-compartment model recognizes the existence of non-equilibrium states during exercise and a slow return to a stationary state (far from equilibrium) during rest. Additionally, the quasi-static approach assumes the whole body to be in equilibrium and therefore considers the body as a single compartment at a homogeneous temperature; that is, there is no distinction between core and skin temperatures in Equation (17), in contrast to Equation (14). 

## 4. Discussion

The present study demonstrates the feasibility of real-time entropy production monitoring in humans exercising under heat stress using a calorimetric approach. Our entropy flow model leads to a resting entropy production rate that is slightly higher (possibly due to the presence of heat stress), yet in good agreement with that of Aoki [5]. However, this value requires repeated and independent measurement without heat stress to be considered a reference standard for resting entropy production. No published results other than those of Aoki are available, which is remarkable given the central nature of entropy production to human life, as publicly recognized by Schrodinger in 1944 [23]. Furthermore, we have investigated the use of classical thermodynamics and the associated entropy change resulting from quasi-static processes. The latter approach highlights the superiority of calorimetry over thermometry to estimate body temperature time series.

### 4.1. On the Relevance of the Entropy Analysis

The entropy flow analysis presented here does not considerably differ from a heat flow approach since body temperature is highly constrained, which is a direct consequence of the narrow range of temperature stability in warm-blooded animals. However, our experimental paradigm offers the possibility of quantifying entropy flows in human bodies and their alteration in association with health, illness and aging, and of further evaluating the possible interpretations of these results and their limitations. In doing so, our entropy analysis allows a deeper explanation of regulatory behavior in novel contexts. For example, a similar entropy flow analysis could be applied to physical systems that do not operate mainly under thermal exchanges, as is the case for the human body, thus offering a unified principle to explain the regulatory behavior of larger classes of physical systems. For physical systems that do not operate under thermal exchanges, entropy flow analysis can still be applied and provide insights into their inner mechanisms; the analysis of these systems is not limited to the use of the first law of thermodynamics, but can be extended to the second law. 

### 4.2. On the Necessity of Considering Non-Equilibrium States

In thermodynamics, the equilibrium state of a system is characterized by its temperature. However, if a participant is resting at a stable temperature and then starts exercising at a high intensity, their body temperature will increase. Once exercise is stopped, and after a sufficiently long cooldown period, the body temperature will eventually decrease to its original value. Since the initial and final temperatures are the same, we could say the body is in the same state, but the participant has depleted most of their energy resources and is exhausted. They would not be able to sustain the same high-intensity exercise; therefore, the initial and final states are different. The most remarkable insight from the theory of thermodynamics is its ability to describe a system in terms of a reduced set of variables, like pressure and temperature. On the other hand, in the study of complex living systems, the human body cannot be similarly reduced to a set of simple thermodynamic variables and, thus, its *state* cannot be properly defined at all-times. Using quantized energy levels to describe the state of systems, namely through the formalism of statistical mechanics, is even less within reach. The definition of a state function implies path independence between two distinct states with different temperatures. For humans, body temperature does not characterize the state of the system because (1) irreversible processes in the body lead to path-dependent transitions and (2) the chemical composition of the body varies during exercise periods.

### 4.3. Non-Equilibrium Steady State: Temperature Gradient of Subcompartments

An alternative approach to estimating the entropy production rate in humans could be to assume that the body is always in a steady state, although a non-equilibrium one. The entropy flow rate could then be calculated from the direct calorimetry measurement of the heat dissipated into the environment divided by the skin temperature and, thus, eliminates the necessity to estimate the metabolic entropy production rate using indirect calorimetry. Given the steady-state approximation, the skin compartment receives and dissipates the same amount of heat from the core; otherwise, skin temperature would not be constant, and the steady-state assumption would not hold. This situation corresponds to a constant heat flow, ${\dot{Q}}_{i}\;$in Equation (12), being transferred across all compartments. Then, the entropy production rates of the core, ${\left(dS_{i}/dt\right)}_{core}$, and that associated with the transfer of heat between the core and the skin region, ${\left(dS_{i}/dt\right)}_{t}$, are given, respectively, by the first and second terms of Equation (12). Interestingly, the interplay of the entropy production rates of the core and the skin systems leads to the following equation
(18)$$\frac{{\left(dS_{i}/dt\right)}_{t}}{{\left(dS_{i}/dt\right)}_{core}}=\frac{T_{c}-T_{s}}{T_{s}}$$
which is mathematically equivalent to evaluating the relative temperature difference between the core and the skin. This relative difference, or more commonly the temperature gradient, between the skin and the core has been studied previously by Cuddy et al. [24]. They have shown that the temperature gradient between the core and the skin during exercise is more indicative of volitional fatigue (i.e., an inability to maintain exercise intensity) compared to the singular measure of core temperature. Although a greater temperature gradient could be seen as a sign of better regulation from a thermodynamics perspective, having a lower skin temperature may not necessarily be beneficial from a physiological perspective. For example, having a lower skin temperature could also mean (1) that you are unable to effectively increase blood flow to the skin, which leads to entropy accumulation within the core, or (2) that you are less efficient at cooling the skin since the rate of sweat evaporation increases with skin temperature [14].

### 4.4. Limitations

Indirect calorimetry estimates internal heat production rates through the expired gases by assuming that they are the product of the oxidation of carbohydrates and, thus, considers that only aerobic metabolism is involved. Anaerobic metabolism, based on the oxidation of lipids, is used mainly when the cardiorespiratory system is unable to provide enough oxygen, that is, for short bursts of activity (e.g., sprinting) at a very high intensity or during prolonged exercise when oxygen requirements are not met [25]. The anaerobic contribution to metabolic heat production was minimized here by keeping the exercise sessions no longer than 30 min (enough to see a difference in entropy rates), and the recovery periods long enough (≥15 min) for the body to replenish its aerobic substrates before the start of the following exercise bout.

One might expect body temperature to be easily measured; however, it is difficult to adequately assess the temperature distribution across the body [17], especially during exercise, when heat is generated non-uniformly. There is in fact a notable increase in activated muscles where the temperature can reach up to 40 °C [14,26,27]. Each organ’s temperature depends on the level of blood perfusion and proximity to heat-generating muscles. Thus, reducing both of the two compartments (core and skin) to single temperature averages represents an obvious limitation. We relied on the work of physiologists and thermal engineers for the best estimation of body temperatures. In particular, modeling the flow between a core compartment and a skin compartment may need adjustments depending on where in the body those compartments are located, e.g., the trunk vs. the head, which entail different sets of tissue composition and thickness. Relatedly, improvements in measuring techniques, both for heat flows and temperature, could allow for the elaboration of a refined multicompartment model where the core (and possibly the skin) region would be further divided into subsystems (e.g., thorax, head, limbs, etc.). While being relatively easy to implement from a mathematical perspective, such an extension to our model would certainly present exceptional experimental challenges associated with measuring these extra variables within a calorimetry chamber in participants performing exercises. We note that similar challenges of modeling scale and tissue heterogeneity, from the meso–macroscopic level all the way down to the nano–meso level of molecular interfaces, arise in other contexts. For example, in biorheology one must choose the biophysical compartments and the linear versus extended Onsager formalism with which to model non-equilibrium thermodynamics [28]. Overall, although the calorimetry method does not provide any information about the temperature distribution, it remains the most accurate method with which to estimate the total amount of heat stored internally from the balance of heat flows. We have showed that thermometry, the alternative approach, leads to inaccurate entropy time series, likely because it provides a highly localized measurement due to the use of a limited number of local probes (typically rectal and/or esophageal) that do not reflect the whole-body temperature or any extrapolation thereof [29]. In the case of a rectal probe, it tends to react quite slowly to changes in core temperature, due to the low heat conductivity of the air that surrounds it in the rectal cavity.

In our approach, the internal entropy production rates and external entropy flows were measured using different physical quantities based on indirect and direct calorimetry, respectively. We specifically chose direct calorimetry as the best method to measure the external heat production of the body, and indirect calorimetry to measure the internal heat production, as they are best possible and available means to do so continuously over time. However, important limitations exist. While the statistical error, presented as the standard deviation in Figure 3 and Figure 4, reflects the biological variability of only 11 subjects, systematic errors may also be present. Systematic errors may be related to the apparatus and measurement techniques that could impact the magnitude and timeline of the computed entropy changes, which were calculated by simultaneously subtracting the rates of entropy production and the entropy flows continuously over time, as is shown in Figure 3. However, years of research with different experimental configurations has led to an understanding and elimination of many of the systematic errors relating to the apparatus shown in Figure 1. For example, this modified Snellen human calorimeter was specifically designed to measure rapid transients in heat loss from an exercising participant with its fast response time (particularly for evaporative heat loss), low thermal inertia, and an unparalleled accuracy of 2.3 W for the measurement of total heat loss [11,13]. Hence, we believe the systematic errors are small and similar across the duration of the experiment and, thus, do not impact our measurements to the extent that they would negate our conclusions about entropy accumulation as a measure of stress. Further experiments could improve the accuracy and reliability of indirect and direct calorimetry as measures of internal and external entropy production, particularly in non-stationary conditions.

### 4.5. Entropy Production in Living Systems

Living systems actively work to maintain the necessary non-equilibrium conditions for metabolic processes to keep running (e.g., chemical gradients for ATP and polymers, and ion gradient across cell membranes). These non-equilibrium conditions can be seen as constraints applied by the system on itself to preserve its structural and dynamical integrity [30]. Although the existence of these self-constraints maintains lower entropy states through a greater microscopic organization, it allows the system to be more efficient at performing work on a macroscopic scale, where thermodynamic entropy flows are possibly maximized. It should be noted that the maximization of entropy production does not imply a maximal generation of energy waste, so long as the energy consumed through metabolic processes enables the living system to achieve greater work output or self-organization. Survival then depends on the ability to make use of an energy source efficiently [31,32]. A system’s capacity for internal order and sustained functions is constrained by its rate of entropy production and entropy flow. Similarly, health and fitness could be described as retaining the ability to maximally increase entropy production when needed. The accumulation of entropy over long periods of time (e.g., years or decades) could be the reason why biological systems lose in efficiency and eventually fail. 

Entropy production is a necessary condition for self-organization (maintenance and healing), while entropy flow is required to prevent entropy accumulation within the body. In this regard, an impairment of entropy production or flow is indicative of a lack of adaptability and, possibly, bad clinical outcomes. On the other hand, it also implies that optimizing the resting and maximal entropy production rates could improve the health status of a patient and their clinical outcome. Therefore, if health is associated with entropy production, numerous additional therapeutic approaches that are not currently considered could be investigated. For example, interventions intended to augment basal or maximal entropy production, the monitoring of entropy production, therapeutic temperature modulation to stimulate entropy production, and the use of entropy production to identify perioperative risks for major surgery all represent separate novel therapeutic options. The accumulation of entropy within the body due to the impairment of entropy flow leads to the concept of entropy as a general measure of stress (e.g., stress entropic load [9]). Strategies to reduce entropy accumulation over prolonged periods of time could become a useful prevention tool to help long-term health and fitness. However, prior to the elaboration of any entropic-based clinical treatment, we first need to better understand how entropy production and entropy flow are affected by different medical conditions (e.g., diabetes) and, more generally, by the inevitable aging process.

## 5. Conclusions

In the present study, we introduced a two-compartment entropy flow model that allows for the real-time monitoring of entropy production in humans exercising under heat stress. Our calorimetric approach provides the much-needed experimental data to further investigate human entropy production in non-equilibrium conditions. This opens new perspectives on the study of fundamental concepts such as health and illness, based on thermodynamic principles and entropy production impairment.

## Figures and Tables

**Figure 1 entropy-25-01290-f001:**
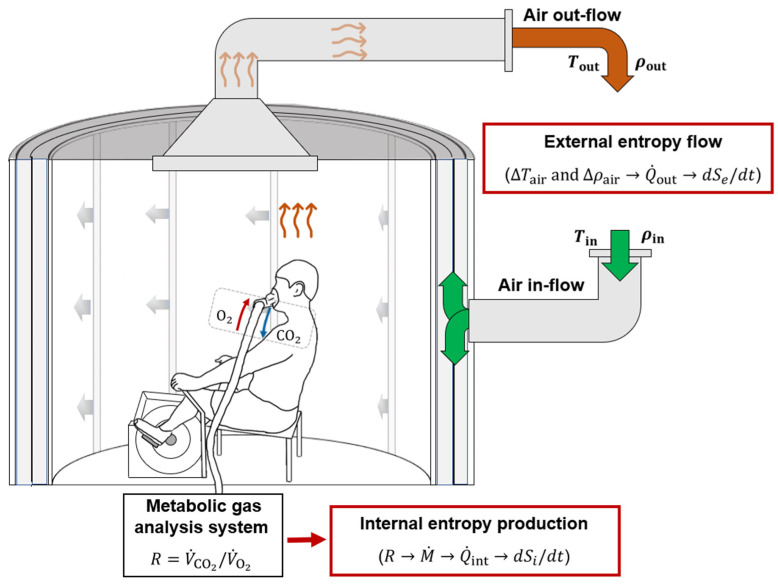
Participants exercise on an upright cycle ergometer within the calorimetry chamber, which essentially acts as a thermal heat bath that maintains a fixed internal temperature, $T_{in}$, and relative humidity, $\rho_{in}$. Internal entropy production is measured using indirect calorimetry; expired gases enter an automated metabolic gas analysis system to measure oxygen uptake (${\dot{V}}_{O_{2}}$) and carbon dioxide production (${\dot{V}}_{{CO}_{2}}$), and determine the respiratory exchange ratio, R, allowing for the calculation of the rates of metabolic energy expenditure, $\dot{M}$, and metabolic heat production, ${\dot{Q}}_{int}$, from Equations (3) and (4), respectively, and, subsequently, the entropy production rate, $dS_{i}/dt$, from Equation (14). External entropy flow, $dS_{e}/dt$, is measured using direct calorimetry; changes in temperature, $\Delta T_{air}$, and relative humidity, $\Delta \rho_{air}$, between the air outflow and inflow, according to Equations (6) and (7), respectively, allow for the measurement of the heat dissipation rate, ${\dot{Q}}_{out}$, and, subsequently, the entropy flow, $dS_{e}/dt$, from Equation (14). Arrows between the variables in the red boxes indicate the logical order of determination up to the calculation of the entropy rates, $dS_{i}/dt$ and $dS_{e}/dt$.

**Figure 2 entropy-25-01290-f002:**
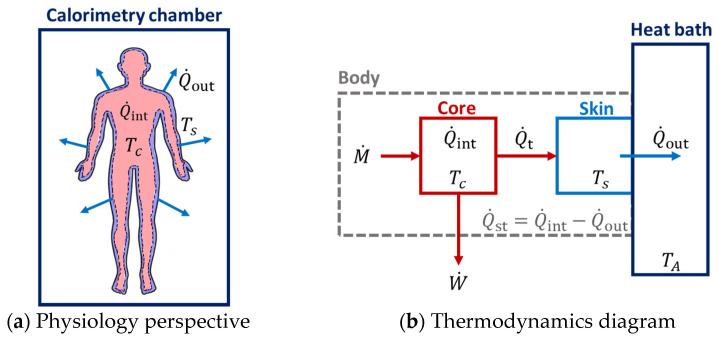
Two-compartment model of entropy flow in humans composed of the core and skin compartments. Entropy is produced throughout the body through irreversible metabolic processes and dissipated to the external environment through the skin. Panel (**a**) shows the physiological viewpoint of heat production and dissipation, while panel (**b**) shows the corresponding thermodynamic diagram. The temperature, $T_{A}$ of the heat bath refers to the ambient temperature inside the calorimetry chamber. The design of the experiment allows for the computation of entropy flows from the heat flows.

**Figure 3 entropy-25-01290-f003:**
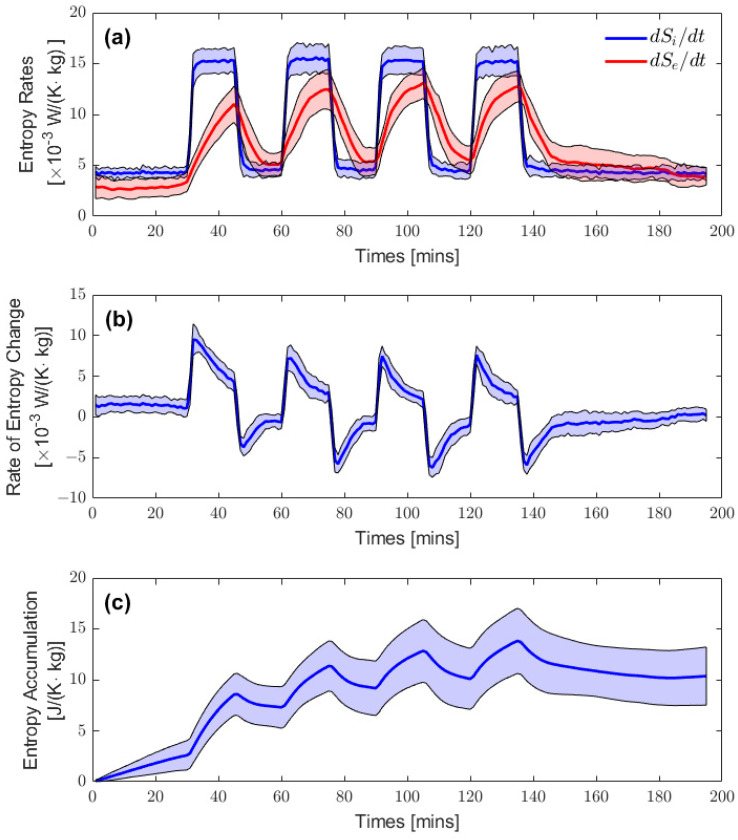
Panel (**a**) shows the average entropy production rate ($dS_{i}/dt$) in blue and the absolute values of entropy flow ($\left|dS_{e}/dt\right|$) in red, as a function of time, calculated using the following equations, $dS_{i}/dt={\dot{Q}}_{int}/T_{c}\;$and $dS_{e}/dt={-\dot{Q}}_{out}/T_{s}$, from Equation (14). Solid lines indicate the average values and the shaded areas indicate the standard deviation among the participants. Panel (**b**) shows the rate of entropy change (dS/dt) that corresponds to the sum of the entropy production and dissipate rates. Panel (**c**) shows the entropy accumulation that corresponds to the cumulative entropy change over time calculated from Equation (15). All the curves presented here are normalized using the participants’ weights prior to computing the shown group averages.

**Figure 4 entropy-25-01290-f004:**
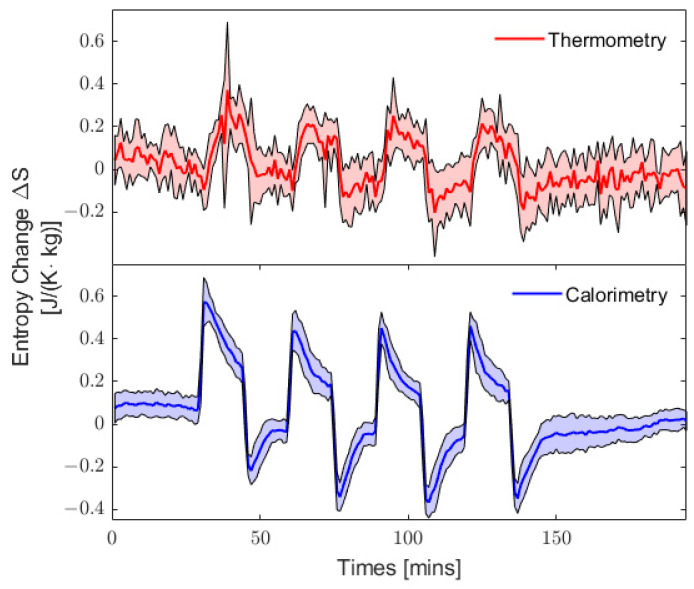
Comparison of the entropy change within the body calculated using Equation (14) with core temperatures estimated from thermometry (in red) and indirect calorimetry (in blue).

**Table 1 entropy-25-01290-t001:** Resting entropy production rate (mean ± SD).

Study	Resting Entropy Production per Unit Body Surface Area (W/K/m^2^)	Resting Entropy Production per Unit Body Mass (×10^−3^ W/K/kg)
This study	0.18 ± 0.01	4.3 ± 0.4
Aoki [5] ^1^	0.172 ± 0.003	3.47

^1^ Aoki’s study measured the basal entropy production of a single middle-aged male. The value reported above corresponds to the average of multiple measurements conducted at different ambient temperatures. The basal value shown above for our study was obtained by averaging the entropy production rate in 11 middle-aged men at a single ambient temperature of 35 ° C. The uncertainty attached reflects the standard deviation over this average.

## Data Availability

De-identified participant data are available from Prof. Kenny (gkenny@uottawa.ca) upon reasonable request and a signed access agreement.

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
