# Peer review of "Continuous Monitoring of Entropy Production and Entropy Flow in Humans Exercising under Heat Stress"

_entropy, 2023, doi:10.3390/e25091290_

Round 1
Reviewer 1 Report
Review of entropy-2525958This could possibly be a useful paper, but it totally misunderstands what entropy is. Entropy is defined in thermodynamics as dS = Q/T where Q is heat exchanged between the system and surroundings and T is the absolute (Kelvin) temperature. In statistical mechanics entropy is defined as S = kBlnW where kB is Boltzmann’s constant and W is the number of energy microstates, i.e. entropy is a summary statistic defining the distribution of energy among the quantized material in the system.
The title, “Continuous monitoring of entropy production and dissipation in humans under heat stress and exercise” is thus a misstatement about entropy. A summary statistic cannot be “produced” or “dissipated”, i.e. entropy can be increased or decreased, but it cannot be “produced” or “dissipated”.
This problem of the language describing entropy continues throughout the paper, e.g. “produce entropy internally and exported to the environment” and “external entropy dissipation” and “internal entropy production (within the body),” and “monitor internal entropy production”.
This statement, “Physical systems generally evolve towards a state of equilibrium characterized by complete homogeneity. This is, in essence, the second law of thermodynamics which states that the increase in entropy manifests itself by the increase in apparent disorder.” is not correct. The second law only deals with the distribution of energy except at 0 K where the entropy of disorder is equal to the entropy of energy distribution. It is a common mistake to equate thermodynamic entropy to disorder.
This statement, “Far from equilibrium, new structures and functions can emerge from the complex interplay of irreversible processes that continuously produce entropy as energy gradients are dissipated.” is also incorrect. “Far from equilibrium” does not necessarily imply “irreversible processes”, irreversibility is a function of the path taken from the non-equilibrium state to equilibrium. Whether or not a process “continuously produces entropy as energy gradients are dissipated” depends on what the “energy gradient” is and whether the process is endothermic, exothermic, or athermic.
“These systems, referred to as dissipative structures, are characterized by self-organized behavior that describes Nature’s tendency to dissipate energy gradients.” Dissipation of an energy gradient doesn’t necessarily create “organization”, it can also create disorder in materials and increase the thermodynamic entropy.
This sentence makes no sense, “In such cases, complex systems can spontaneously form to feed upon these energy gradients, adopting complex (i.e., highly ordered) geometries that are impossible under equilibrium conditions.” Complex systems may be organized, but they cannot have highly ordered geometries.
“Complex living systems such as the human body can be regarded as open thermodynamical machines that inherently transform energy sources mostly into heat in order to produce a relatively small portion of useful work.” Living systems can be just as easily defined as closed systems by including material inputs and outputs within the system. The boundary of a system is arbitrary and usually defined so as to simplify the thermodynamic calculations and measurements. Defining living systems as open systems, particularly in this study, is an unnecessary complication. “Useful work” is defined in thermodynamics as work done on or by the system on the surroundings, and as such always involves an equivalent exchange of energy with the surroundings. In this study, no work is done on the surroundings, only heat, O2 and CO2 are exchanged between the system (which I assume is the human body since it is never really defined in the paper) and the surroundings.
The statement, “Here we develop an experimental-theoretical framework to understand heat and entropy flows in healthy humans that alternate work and rest periods.” is incorrect, there is only heat flow, entropy does not flow.
This phrase has the same problem, “entropy exchanged with the surroundings”, entropy is not a “material thing” that can be moved around.
“It has been hypothesized that nonlinear systems with many degrees of freedom,” Degrees of freedom in what variable? “like the human body, allow for the existence of multiple steady states” Steady states of what process? “with the associated increase in stability” Of what? “when the system tends to one of these states after perturbation [3].”
Same problem, “heat and entropy flows”, heat flows but entropy doesn’t.
“The thermodynamical definition of entropy will be applied to this experiment, as opposed to the informational entropy derived from information theory.” This is good, but the authors failed to recognize the implications in the rest of the paper.
The statement, “Although the entropy content of the human body cannot be measured,” is wrong because the entropy (not “entropy content”) of a human body could be measured. “entropy flows” No it doesn’t. “and their imbalance are quantifiable.”
“?̇??? is the rate of metabolic heat production and ? ̇ is the external work rate performed by the participants on the cycle ergometer.” No thermodynamic work is done in this system, there is only an increase in metabolic heat production during exercise.
“chemical energy is needed to provide this internal work.” There is no such thing as “internal work” defined in thermodynamics. There is a definition of “internal work” as the energy necessary to separate particles in a system that doesn’t apply to this study.
“The rate of metabolic energy expenditure ?” Metabolic heat rate? ̇ “was estimated spirometrically from a known respiratory exchange ratio (RER, here equal to 1) between the rate of carbon dioxide production ?̇??? and the rate of oxygen consumption ?̇? ? [15],” Equation 4 needs more explanation. If RER = 1, the first term = ec, and the second term =0.
“Here ?? is the caloric equivalent per liter of oxygen for the oxidation of carbohydrates and ?? is the caloric equivalent per liter of oxygen for the oxidation of fat.” What values were used for ec and ef?
“The “external work rate ?” (use different terminology or delete this phrase) “was continuously adjusted by changing the cycling resistance during the exercise periods to ensure the targeted metabolic heat production of 400 W remained constant in time.”
“The typical values for work rates were in the range of 70 W, which represents a moderate to high intensity exercise.” Is this part of the 400 W stated above?
Many people would disagree with the statement that, “Indirect calorimetry is widely accepted as the most accurate method of measuring the rates of metabolic energy expenditure ? ̇ and heat production ?̇??? in humans;” Also, “metabolic energy expenditure” and “heat production” are equivalent, not two different things.
Same problem, “entropy flows from the heat flows”. Entropy is not a material thing so it can’t flow.
“We restricted the entropy rate contributions to real-time heat production” Entropy doesn’t contribute to heat production. “because the human body is inefficient at converting metabolic energy into external work.” This statement requires a lot more nuance. How is efficiency defined? How is metabolic energy defined? What kind of external work? Living organisms are not heat engines, so there has never been a satisfactory definition of efficiency.
“normalized by BSA or mass” BSA is not defined.
“(x10-3 W/K/kg)” Does this mean kW/kg? if so then use kW. Also entropy is not defined as a rate, use “the rate of entropy increase”.

Reviewer 2 Report
The authors apply the fundamental laws of thermodynamics to examine an open biological system. For my part, I like the fact that, in addition to physical-chemical problems, concepts related to irreversibility and dissipation also appear in other fields of science. Later, it would be worth extending the model to cases with variable temperature measurements, and it would be exciting to consider radiation processes. The current results refer to stationary, quasi-stationary cases for which the mathematics applied is sufficient. You have to jump a level later.
Note: The title of the journal is missing in the cited [6]: Journal of Theoretical Biology.
Reviewer 3 Report
In the study of the Canadian Group, a two-compartment entropy flow model for the continuous monitoring of entropy production in humans evaluated during physical exercise under heat stress has been presented successfully.
A modified Snellen calorimeter, allowing for both indirect (metabolism-oriented) and direct (temperature sensitive) has been used in order to estimate thoroughly the entropy production rates inside and outside a human subject. The perennially alive conception of calorimetry makes direct use of the 1st law of thermodynamics, and its basic and/or fundamental theoretical expression is presented in terms of the rates of entropy change, as well as how quickly the heat is dissipated as well as what is the power associated to such a dissipation.
Furthermore, the presented discussion, and the results exemplifying it, are also presented in terms of the (so important) stationary states, giving rise to a well-justified opinion that non-stationary states, characteristic of nonlinear (Prigogine-type) entropy production are of importance for the human subjects. This is also due to the biophysical fact that a living system cannot live at all in a simple exponentially vanishing stationary state of Gibbs-Boltzmann type.
The paper is sufficiently well-written and develops its core discussion in an open-minded, non-arrogant (or, obscure) style, provoking this way a sufficiently sensitive reader to a type of dialogue; this is very good for the paper.
The present reviewer can find certain minor drawbacks of the presentation. At line 292, after Eq. (14) one may spot "expressed in per second" which should necessarily be amended. Methodologically, over the almost full course of the presentation, one may find an expression like 'entropy dissipation' which is meant as the entropy disintegration or de-accumulation. But consequently, the expression of entropy dissipation is used: a classically oriented reader has heard about energy dissipation such as that manifested by heat dissipation, or the like, but the mentioned expression is non-standard and should concisely be explained to the reader in advance.
Next, when looking at Eq. (18) one would be keen to interpret it in terms of a neg-efficiency since its l.h.s. equals: -(1-T_c/T_s) - am I right, and whether a certain associated and possible interpretation of the two-compartment "engine" in terms of -(1-T_c/T_s) might be instructive/useful?
Further, possibly the most important, the two-compartment conception in terms of core and skin outcomes has its interesting shortages, as the Authors tried to discuss during their presentation. The shortage is, of course, not only the gradient-like (linear) analysis but also this all which belong to the active matter staff. On the one hand, imagine as the core the nervous cellular system and as the skin that of our heads. On the other hand, the articular cartilage, envisaged as the core, is an anervous (and, avascular) system, adjacent to the knee's skin, for example. It makes a difference in assuming what the core, and possibly what the skin can be, see but J. Phys. D: Appl. Phys. 55, 483002 (2022) DOI 10.1088/1361-6463/ac90d1 for comprehending the reviewer's reasoning. Thus, the discussion should be prolonged by emphasizing what any differentiation can cause, just for the completeness of this valuable study.
Last, the so relevant references Nos. 5 and 6 are incomplete, and they should be complemented by: the second author and name of the journal (not mentioning the flawed link), respectively; thus, all references have to be checked carefully once again.
To summarize, the paper is clearly publishable after the minor revision to be made by the Canadian Group.
Round 2
Reviewer 1 Report
This paper is based on a fundamentally flawed model. Living systems are organized, but not ordered in the crystallographic sense assumed by the third law. Thermodynamic entropy refers to the Boltzmann distribution of quantized energy, not the distribution of matter except at zero Kelvin. Measurements and calculations of the Gibbs energies, enthalpies and entropies of the matter in living organisms shows it is not in a high energy metastable state with respect to totally random matter of the same elemental composition. Thus, there is no "energy gradient" that drives organization, a fundamental assumption of the model. The authors correctly state that entropy is not conserved, but assume a conservation law in the basic equation in their model, i.e., dStotal = dSinternal + dSexternal.
The study measured the rates of oxygen consumption, CO2 production, and the rate of substrate combustion as a function of the rate of exercise which is interesting and could make a good paper if they simply treat the human body as a catalyst that allowed the reaction to happen. The application of non-equilibrium thermodynamics is an unnecessary complication that obscures any meaningful results.
Reviewer 3 Report
Only a minor improvement in the end of p. 15 and beginning of p. 16.
Namely, instead of the term nonlinear Onsager put modified or extended Onsager.
Then on p. 16 remains the... instead of remains to should appear. Overall, the notion of entropy flow is suitable.
Author Response
We would like to thank the Reviewer for his time and his constructive criticisms that improved the quality and readability of our manuscript. Please see the attachment for the revisions made according to their suggestions.

Round 3
Reviewer 1 Report
Please see the attached file.
